# Self-Management Mobile Virtual Reality Program for Women with Gestational Diabetes

**DOI:** 10.3390/ijerph18041539

**Published:** 2021-02-05

**Authors:** Sung-Hoon Kim, Hye Jin Kim, Gisoo Shin

**Affiliations:** 1Department of Internal Medicine, Yonsei University Wonju College of Medicine, Wonju 26493, Korea; shkim83@yonsei.ac.kr; 2Department of Nursing, Kyungsung University, Busan 48434, Korea; khj325@ks.ac.kr; 3College of Nursing, Chung-Ang University, Seoul 06974, Korea

**Keywords:** women, virtual reality, self-management, health promotion

## Abstract

The incidence of type 2 diabetes and gestational diabetes shows an increasing trend worldwide, and women diagnosed with gestational diabetes are more likely to develop type 2 diabetes within 5 years of delivery. This is closely related to lifestyle habits such as dietary intake and physical activity; hence, self-management should be continuously practiced. However, after childbirth, women find it challenging to practice self-management due to physical discomfort and child rearing. Therefore, it is necessary to develop an intervention program that is tailored to the characteristics of each participant and allows them to practice self-health management efficiently without time and space restrictions. This study aimed to develop a self-management mobile virtual reality program and investigate its efficacy in preventing type 2 diabetes after childbirth among women diagnosed with gestational diabetes. Intervention with the self-management mobile virtual reality program was performed for 12 weeks. The data of 57 participants in the experimental group and 62 participants in the control group were analyzed. After 12 weeks, the body weight, body fat, hemoglobin A1c, and fasting glucose were decreased in the experimental group compared with the control group. In addition, the dietary habits and health-promoting lifestyle profile were improved in the experimental group compared with the control group. These findings demonstrated that a self-managed mobile virtual reality program could be used as an intervention method for health promotion. To verify the effectiveness of intervention with the self-management mobile virtual reality program, a follow-up study with a larger number of research subjects should be conducted in the future.

## 1. Introduction

Gestational diabetes mellitus (GDM) is diagnosed based on the results of an oral glucose tolerance test conducted between 24 and 28 weeks of pregnancy for women without a history of diabetes [1,2,3]. In recent years, the prevalence of GDM is gradually increasing in Korea and around the world as the prevalence of obesity increases, and the age of pregnancy increases along with the delay of marriage among young adults [4]. Pregnant women with gestational diabetes may have complications such as gestational hypertension, increased frequency of cesarean section, damage during delivery, and dystocia; however, they are a high-risk group with a high incidence of type 2 diabetes mellitus even after childbirth, and hence continuous management is necessary [5]. According to the results of a previous study that followed GDM women for 5 years, around 10% of cases of GDM to type 2 diabetes progression occurred immediately after delivery. More than 30% of women had type 2 diabetes within 1 year after delivery, and subsequently, 5% of them developed type 2 diabetes every year. Five years after childbirth, around 50–60% of women with gestational diabetes developed type 2 diabetes [6]. 

GDM etiology is related to pancreatic beta-cell dysfunction and marked insulin resistance secondary to placental hormonal release [7]. Insulin resistance increases toward the end of pregnancy, and the increased insulin resistance facilitates the transport of glucose to the placenta for fetal growth; however, this can cause gestational diabetes due to high blood glucose levels. The etiology of gestational diabetes is unclear. Nevertheless, when insulin resistance increases during pregnancy, the beta-cell function of the pancreas increases to compensate, and it has been suggested that if this compensatory beta-cell function is not sufficient, gestational diabetes may develop [8]. The prevalence of gestational diabetes varies according to ethnicity, age, family history of diabetes or obesity, and individual lifestyle, and it has been reported to affect 10–20% of all pregnant women worldwide [9]. In addition, GDM is closely related to lifestyle habits such as dietary intake and physical activity. According to the results of a study, for all stages of type 2 diabetes, when interventions for lifestyle correction with diet and exercise therapy were provided, the incidence of type 2 diabetes was decreased by 58% after 2 years, and the risk factors for diabetes continued to decline after 7 years of follow-up [10]. The International Diabetes Federation recommends weight maintenance before pregnancy and within three months after delivery to prevent type 2 diabetes in women with gestational diabetes; thus, improving lifestyle habits and self-management strategies such as diet and exercise are essential [11].

### 1.1. Physical and Psychological Characteristics of GDM Women after Childbirth

During pregnancy and delivery, a long journey of around 280 days, women experience many changes both physically and psychologically [12]. In particular, as women after childbirth experience mental and physical fatigue, breast engorgement, episiotomy or physical discomfort to the cesarean section, rapid changes in hormones, and role changes due to new circumstances, they sometimes experience loss of self-esteem and poor quality of life [13]. Moreover, when women with gestational diabetes are presented with an unexpected GDM diagnosis, they feel embarrassed, depressed, and anxious, which puts them in a physically and psychologically debilitated state not only during pregnancy but also after childbirth [14]. Women diagnosed with GDM may be compelled to adopt a healthy lifestyle (e.g., healthy diet and regular exercise) by completely changing their regular lifestyle habits (e.g., familiar foods that were consumed before pregnancy and even after childbirth). Moreover, because they need to monitor factors affecting type 2 diabetes development by undergoing periodic blood glucose tests, women with gestational diabetes have more challenges to cope with after childbirth than those who do not have gestational diabetes [15]. 

Although stress levels are known to increase in women with gestational diabetes after childbirth, current research and interventions targeting women with gestational diabetes have mainly focus on pregnancy, and studies involving interventions after childbirth remain insufficient [16]. Consequently, adequate interventions are limited despite the awareness that gestational diabetes can progress to type 2 diabetes after childbirth [17]. Therefore, modifying health behavior to prevent type 2 diabetes after childbirth among women with gestational diabetes is important. For this purpose, intervention after childbirth is necessary for women with gestational diabetes, which could help them practice self-management successfully according to their needs after childbirth.

### 1.2. Self-Management Mobile VR Program

Virtual reality (VR) was developed by the United States Aeronautics and Space Administration (Washington, DC, USA) to create a visual virtual environment for astronaut training, and since the 2000s, it has been developed in three dimensions (computer-based or mobile-based) to achieve an immersive experience [18]. VR intends to create an environment similar to reality with the use of state-of-the-art technologies (e.g., computers and graphics) by directly interacting with the five senses (e.g., sight, touch, and smell) to provide realistic spatial and temporal experiences [19]. 

VR simulation programs are increasingly used in the field of airplane maneuvering or combat, and they have been applied in various forms in games, sports, architecture, engineering, medicine, and nursing [18]. With the recent spread of COVID-19, VR programs linked to smartphone mobiles have attracted great interest worldwide [20]. In a given VR environment, subjects are provided with an independent, immersive virtual space that is separated from their surroundings. It has the advantage of not being spatially or temporally limited, and repeated learning or training is possible according to the subjects’ convenience [18]. In addition, because the behavioral information of the subjects can be stored on a mobile, it is possible to construct a system that can analyze the subjects’ behavior and provide feedback [20].

Thus far, studies using VR programs for health problems have mainly aimed to support rehabilitation. The effectiveness of VR programs has been reported in an intervention study aiming to restore the motor function and physical function of patients with brain damage, stroke patients, and adults with disabilities [21]. Interventions using a mobile VR self-management program may be useful for the postpartum management of women with gestational diabetes because after childbirth, women are physically weak and may have difficulties in visiting the hospital due to parenting obligations [22]. Moreover, although awareness of the importance of prenatal care is high, the awareness of the importance of postpartum care is low [23]. Therefore, in order to prevent type 2 diabetes after childbirth among women with gestational diabetes, it is necessary to develop an exercise program that can be used regardless of time and place with a self-regulating diet regime, which can relieve stressors such as parenting stress.

The present study investigated the efficacy of intervention using a self-management mobile VR program (mobile VR program), which may be useful for the postpartum management of gestational diabetic women considering that women are physically weak after childbirth and have difficulties in visiting the hospital due to their responsibility of raising children. The hypotheses of this study were as follows:

**Hypothesis** **1.**
*Mobile VR program may have positive effects on physiological variables.*


**Hypothesis** **2.**
*Mobile VR program may be associated with a higher level of diabetes knowledge.*


**Hypothesis** **3.**
*Mobile VR program may have positive effects on eating habits.*


**Hypothesis** **4.**
*Mobile VR program may be associated with a better health-promoting lifestyle profile.*


**Hypothesis** **5.**
*Mobile VR program may be associated with lower parenting stress.*


## 2. Methods

### 2.1. Research Design and Participants

This was a quasi-experimental study that investigated the effects of mobile VR intervention on women with gestational diabetes (Figure 1). In this study, women who visited three hospitals in South Korea from May 2018 to February 2019 were selected. A total of 64 participants who met the selection criteria were selected as the experimental group, and the selection criteria were as follows: (1) women aged >20 years who gave birth after being diagnosed with gestational diabetes, and (2) women who had no visual, auditory, or active disabilities in using the mobile VR program. In addition, 64 women were selected in the control group by matching for age, birth experience, type of birth, family history of diabetes mellitus, and breastfeeding after birth after completing the baseline measures for the experimental group. At the end of the mobile VR program intervention, the data from 57 participants in the experimental group and 62 participants in the control group were used for the final analysis, excluding the data from 9 participants who withdrew their intention to participate in the study due to personal reasons. Seven days after intervention with the self-management mobile VR program (13 weeks after childbirth), participants in both the experimental group and the control group were asked to visit the hospital, and their body weight, body fat, hemoglobin A1c, and fasting glucose were measured. They filled out a questionnaire on their diabetes knowledge, dietary habits, health-promoting lifestyle profile, and parenting stress.

### 2.2. Mobile VR Program Development

We developed the mobile VR program on Android (Figure 2A). The mobile VR program consisted of an exercise program, a diet/nutrition program, laughter therapy for stress relief, and a neonatal first aid program. 

The exercise program included 123 types of exercises ranging from light stretching to strength training that can be selected and performed by the participants (Figure 2B), and we designed the program by capturing the motion of an actual fitness trainer (Figure 2C). The diet/nutrition program recorded the dietary intake of the participants based on a database of 19,700 types of Korean food. The adequacy of dietary intake, including the adequacy of calories, could be evaluated through the use of graphs and scores according to the BMI (Body mass index) of each participant (Figure 2D). The neonatal first aid program provides step-by-step guidance on what to do in cases of neonatal vomiting, nasal obstruction, infant colic, cramps, fever, and hiccups (Figure 2E). There were 18 types of laughter therapy exercises for stress relief, and each element required approximately 2 min and 30 s. This program was developed using a 360° rotation technique to increase the 3D effect, which allowed deep breathing exercise (abdominal breathing) (Figure 2F). 

In this study, two exercise trainers, two clinical dietitians, two diet experts, two neonatal specialists, two nurses, and three laughter therapists participated in the development of the self-management mobile VR program. 

Moreover, during the program development process, we received advice from gynecologists, sports medicine specialists and psychiatrists, and three professors in the nursing department. The developed mobile VR program was registered with the Korea Copyright Commission for copyright protection.

### 2.3. Experimental Intervention

The participants assigned to the experimental group were asked to download and install the mobile VR program on a mobile phone the day before the scheduled delivery date or the day after delivery. Moreover, we provided VR headsets necessary for the mobile VR program as well as gave a presentation on how to use this program with supporting educational content. In the hospital, the participants assigned to the control group were provided with written educational materials on GDM in pregnant women, and the physiological variables of both the experimental group and the control group were measured before intervention. Mobile VR intervention took place from May 2018 to February 2019, and the intervention period was 12 weeks after childbirth. Seven days after the mobile VR program intervention (13 weeks after childbirth), postpartum physiological variables were measured. After the completion of the experimental intervention in this study, the participants in the control cohort were also offered the chance to use the mobile VR program.

### 2.4. Measurements

Physiological variables. The physiological variables measured in this study were body weight (kg), body fat (%), fasting glucose (mg/L), and HbA1c (%). The reliability was Cronbach’s α = 0.89.

Diabetes knowledge. The level of diabetes knowledge was evaluated with 24 items [24]; the higher the score, the higher the knowledge level (0 or 1 point). The reliability was Cronbach’s α = 0.86.

Dietary habits. The assessment of dietary habits was performed with 20 items rated on a 5-point scale based on the dietary assessment [25] of Korean adults selected from the Korean National Health and Nutrition Examination Survey conducted by the Korean Centers for Disease Control and Prevention; the higher the score, the higher the quality of eating habits. In this study, the reliability was Cronbach’s α = 0.82.

Health-promoting lifestyle profile. The health-promoting lifestyle profile was evaluated with 52 items on a 4-point scale [26]; the higher the score, the better the health-promoting lifestyle profile. In this study, the reliability was Cronbach’s α = 0.90.

Parenting stress. Parenting stress was evaluated with 23 items on a 5-point scale [27]; the lower the score, the lower the parenting-related stress. The reliability was Cronbach’s α = 0.88. 

### 2.5. Ethical Considerations

This study was conducted after obtaining approval from the Institutional Review Board of Women’s Medical Center (CGH-IRB-2017-44). After explaining the purpose of the study, confidentiality of the data, and disposal of the data after study completion, we received written consent for voluntary participation from our study participants. The participants were informed that they can withdraw from the study at any time and that there were no penalties if participation was withdrawn. Moreover, they were informed that the collected research data will not be used for any purpose other than for this research and will be discarded according to the Bioethics Act.

### 2.6. Data Analysis

The data were analyzed using the SPSS 26.0 program (IBM Corporation, Armonk, NY, USA). The participants’ general characteristics, physiological variables, diabetes knowledge, dietary habits, health-promoting lifestyle profile, and parenting stress were analyzed using the frequency, percentage, mean, and standard deviation. In addition, *t*-test or chi-square test was performed for each sociodemographic and baseline variable to ensure the equivalence of groups. Furthermore, the difference in the intervention effect between the experimental group and the control group was analyzed by independent *t*-test.

## 3. Results

### 3.1. General Characteristics of the Participants

Among the participants in this study, the number of older women aged 35 years or older was 62 (52.1%), and the number of primiparous women was 80 (67.2%). In addition, 72 women (60.5%) had normal vaginal delivery, and 35 women (29.4%) had a family history of diabetes (Table 1).

### 3.2. Homogeneity Test between the Experimental Group and the Control Group

Based on the results of the homogeneity test between the two groups for the dependent variables measured in the preliminary survey, there was no difference in the body weight, body fat, fasting glucose, hemoglobin A1c, diabetes knowledge, dietary habits, health-promoting lifestyle profile, and parenting stress. Thus, the two groups were found to be homogeneous (Table 2).

### 3.3. Intervention Outcomes in the Experimental Group and the Control Group

The outcomes in the experimental group and the control group following intervention with the mobile VR program for 12 weeks are shown in Table 3, and the results of hypothesis testing were as follows:(1)Following intervention with the mobile VR program, the body weight (−5.65 ± 12.90), body fat (−2.52 ± 7.09), fasting glucose (−3.26 ± 16.88), and hemoglobin A1c (−0.18 ± 0.38) were significantly lower in the experimental group than in the control group; thus, hypothesis 1 was supported.(2)Following intervention with the mobile VR program, the level of diabetes knowledge was increased in both the experimental group (0.11 ± 0.90) and the control group (0.10 ± 0.09); however, hypothesis 2 was rejected as there was no statistically significant difference between the two groups.(3)Following intervention with mobile VR program, the score for dietary habits was significantly higher in the experimental group (0.74 ± 0.63) than in the control group (0.34 ± 0.58); thus, hypothesis 3 was supported.(4)Following intervention with the mobile VR program, the total mean score of health-promoting lifestyle profile was significantly higher in the experimental group (0.34 ± 0.49) than in the control group (0.18 ± 0.39); thus, hypothesis 4 was supported.(5)Following intervention with the mobile VR program, parenting stress was increased in both the experimental group (0.02 ± 0.60) and the control group (0.10 ± 0.81), and there was no statistically significant difference between the two groups; thus, hypothesis 5 was rejected.

## 4. Discussion

Here, we developed a mobile VR program for women diagnosed with GDM and used it for 12 weeks to determine its efficacy. As the mobile VR program developed in this study provides auditory and visual feedback according to the participants’ behavior, the participants were assigned a health trainer, clinical nutritionist, and medical staff to aid their health care process. Therefore, the overall exercise routine in the experimental group lasted for an average of 20 min with 2.5 times per day (range, 2–6 times), and the dietary recording was used 1.8 times per day (range, 1–5 times). After 12 weeks of mobile VR intervention, the physiological variables (body weight, body fat, fasting blood glucose, and hemoglobin A1c) in the experimental group were significantly different compared with the variables in the control group. These results are consistent with those of a previous study showing that regular exercise and lifestyle regulation centered on a proper diet are important elements for type 2 diabetes treatment; additionally, these elements are meaningful because they have a positive effect [28]. An interesting finding from another study was that exercising three times a week for 40 to 60 min improves blood sugar control in GDM women and reduces type 2 diabetes after childbirth with GDM [29]. In addition, age and family history of type 2 diabetes do not undermine the improvement in insulin sensitivity caused by exercise because exercise capacity occurs in parallel with improvement in muscle strength and body composition. These results highlight the importance of promoting exercise as a way to prevent type 2 diabetes [30]. Therefore, mobile VR exercise intervention, specifically looking at the effects on strategies to improve insulin intolerance, should be explored to maximize the benefits for GDM women after childbirth.

The meal quality of the participants in the experimental group was improved in this study. The mobile VR program showed the calories, carbohydrates, proteins, fats, and essential nutrients to be consumed per day for each subject and indicated visually whether they met the nutrient requirements according to their dietary record. This finding is in line with a previous study; for example, in the study of motivation enhancement for obesity patients in South Korea [31], it showed that the dietary pattern may change depending on dietary recording. In addition, dietary recording is effective in achieving weight loss and preventing the metabolic diseases by correcting dietary habits and improving the quality of meals [32]. 

According to the health-promoting lifestyle profile of the experimental group in this study, there were statistically significant differences in the scores of physical activity, nutrition, spiritual growth, and interpersonal relation. Reasonable lifestyle habits may have a positive effect [33]. Evidence is rapidly accumulating that postpartum behavior can be important in preventing long-term progression to diabetes, but lifestyle changes can be difficult to implement in the critical year after childbirth, because postpartum women may face pressure to take care of a new baby [34]. Moreover, another study found low-risk perception for future type 2 diabetes and suboptimal levels of physical activity and a lack of knowledge about the necessary lifestyle modifications [35]. As a promising strategy to tackle these challenges, mobile VR interventions along with a health-promoting lifestyle should be offered to women after childbirth with GDM.

However, although the level of diabetes knowledge was increased in both the experimental group and the control group, the results were not statistically different. It is possible that the hospital has provided sufficient education orally or in writing about diabetes to women diagnosed with GDM. Additionally, GDM can cause complications in the fetus, such as intrauterine fetal death, macrosomia, and fetal growth restriction, and the women may be anxious about the health of their fetus [1,2,3]. Thus, they may have relied on the internet and social networks to obtain the relevant medical knowledge [36].

Furthermore, there was no significant difference in parenting-related stress between the two groups. These findings are in agreement with those of a previous study, which revealed that the biggest concerns of Korean parents with young children these days were parenting, low birth rates, and avoidance of marriage by South Korean women [37]. In addition, South Korean women are usually anxious about employment and housing, and in the case of women who are working, it has been reported that the stress of parenting after childbirth is increased due to the notion that they could not continue to work and fulfill parenting responsibilities at the same time [38]. This parenting stress is also reportedly correlated with postpartum depression, postpartum evasion, and maternal attachment disorder in women [39]. To alleviate the parenting stress of women after childbirth, various measures should be taken at the national and social levels, and the effect of mobile VR intervention on parenting stress should be clarified in further studies using representative samples.

This study has few limitations. First, the mobile VR program developed in this study cannot run on the iOS system (iPhone) with a VR device (headset and gear), because it is developed on the Android operating system. Moreover, to date, both Android and iOS have limitations with regard to running VR applications; hence, this issue needs to be considered during intervention. Second, VR can usually cause cybersickness. As most VR devices are developed to continuously change the field of view using 3D or 360° techniques to improve the immersive experience, motion sickness syndromes such as car sickness, sea sickness, and air sickness may occur [40]. When using the VR program, participants must be careful not to exceed one hour in a stretch and should take a 10 min break between sessions. Lastly, this study was a quasi-experimental study, which has limitations in verifying the causal relationship between the participants’ characteristics such as age and family history of diabetes and the results of intervention with the mobile VR program. Therefore, in the future, a follow-up study that considers the factors affecting the prevalence of type 2 diabetes after childbirth among women diagnosed with gestational diabetes should be conducted. 

## 5. Conclusions

In this study, a self-management mobile VR program was developed for women who gave birth after being diagnosed with gestational diabetes, and its effectiveness was evaluated. The mobile VR program consisted of an exercise program, a diet program, laughter therapy, and a neonatal first aid program, and 64 participants in the experimental group used it for 12 weeks. The physiological variables, dietary habits, and health-promoting lifestyle profile of the experimental group showed statistically significant improvements compared with those of the control group. Although further long-term studies with higher-level evidence on the effectiveness of self-management mobile VR program intervention are needed, this study is meaningful in that it provided preliminary data to support self-management through a VR program. 

## Figures and Tables

**Figure 1 ijerph-18-01539-f001:**
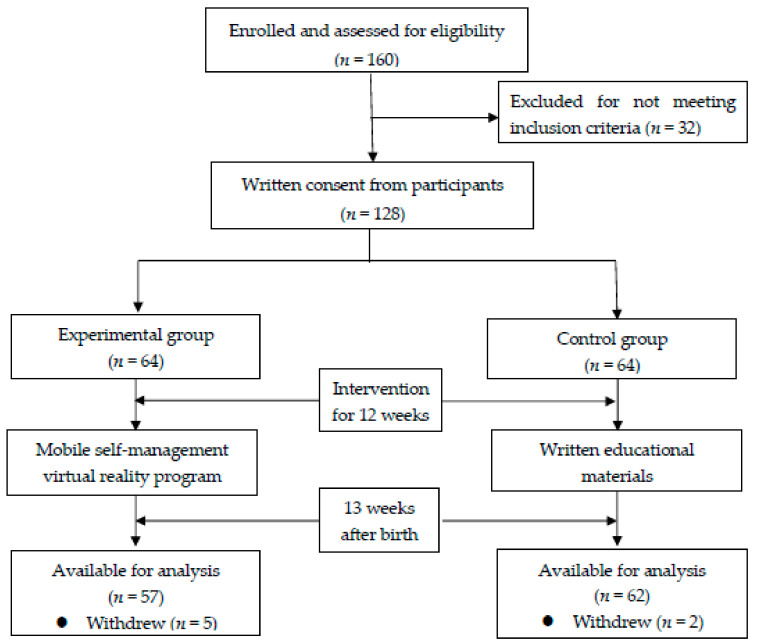
Flow diagram of the study.

**Figure 2 ijerph-18-01539-f002:**
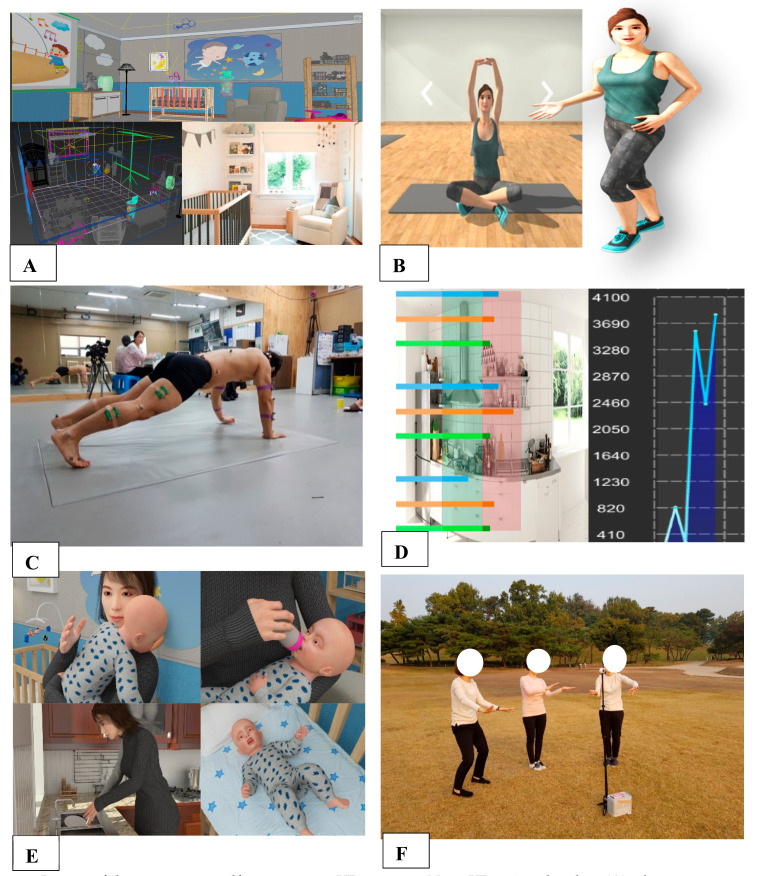
Images of the postpartum self-management VR program. Note: VR = virtual reality; (**A**) = home screen on Android; (**B**) = a type of exercise in the exercise program; (**C**) = capture of the movement/motion of an actual fitness trainer with the recording of the trainer’s voice; (**D**) = evaluation of the participants’ diet based on the body mass index; (**E**) = neonatal first aid program; (**F**) = laughter therapy using 3D or 360° techniques for stress relief.

**Table 1 ijerph-18-01539-t001:** General characteristics of the participants.

Categories	Total (*n* = 119)	Experimental Group (*n* = 57)	Control Group (*n* = 62)	Χ^2^/*t*-test	*p*
	*n* (%)/Mean ± SD				
Age (years)				1.59	0.158
Over 35	57 (47.90)	26 (45.61)	31 (50.0)		
Below 35	62 (52.10)	31 (54.39)	31 (50.0)		
Birth experience				2.99	0.084
Primipara	80 (67.23)	38 (66.67)	42 (67.74)		
Multipara	39 (32.77)	19 (33.33)	20 (32.26)		
Type of birth				0.57	0.449
Normal vaginal birth	72 (60.50)	34 (59.45)	38 (61.28)		
Cesarean section birth	47 (39.50)	23 (40.35)	24 (38.71)		
Family history of diabetes mellitus				0.78	0.675
Yes	35 (29.41)	16 (28.07)	19 (30.65)		
No	84 (70.59)	41 (71.93)	43 (69.35)		
Breastfeeding after birth				2.13	1.664
Yes	111 (93.28)	54 (94.73)	57 (91.94)		
No ^↑^	8 (6.72)	3 (5.26)	5 (8.06)		

^↑^ Fisher’s exact test.

**Table 2 ijerph-18-01539-t002:** Homogeneity test between the experimental group and the control group.

Categories		Experimental Group (*n* = 57)	Control Group (*n* = 62)	*t*-test	*p*
		Mean ± SD			
Physiological variables	Body weight (kg)	67.15 ± 12.95	69.91 ± 12.58	1.18	0.24
	Body fat (%)	34.52 ± 4.48	35.94 ± 6.26	1.42	0.157
	Fasting glucose (mg/L)	96.00 ± 16.46	101.61 ± 16.32	1.87	0.064
	HbA1c (%)	5.56 ± 0.36	5.55 ± 0.36	0.92	0.362
Diabetes knowledge		0.54 ± 0.93	0.52 ± 0.84	−1.11	0.271
Dietary habits		3.32 ± 0.54	3.45 ± 0.57	1.26	0.211
Health-promoting lifestyle profile	Health responsibility	2.77 ± 0.80	2.60 ± 0.75	−1.25	0.295
	Physical activity	2.61 ± 0.80	2.56 ± 0.68	−0.38	0.707
	Nutrition	2.26 ± 0.77	2.37 ± 0.77	0.78	0.436
	Spiritual growth	2.24 ± 0.80	2.23 ± 0.73	−0.99	0.922
	Interpersonal relationship	2.37 ± 0.79	2.25 ± 0.81	−0.85	0.395
	Stress management	2.61 ± 0.73	2.74 ± 0.58	1.02	0.312
	Total	2.48 ± 0.37	2.46 ± 0.37	−0.33	0.743
Parenting stress		3.27 ± 0.50	3.42 ± 0.39	1.79	0.077

**Table 3 ijerph-18-01539-t003:** Intervention outcomes in the experimental group and the control group.

Categories		Experimental Group		Control Group		*t*-test *(P)*
		(*n* = 57)		(*n* = 62)		
		Follow-Up	Mean Differences (Post-Baseline)	Follow-Up	Mean Differences (Post-Baseline)	
		Mean ± SD		Mean ± SD		
Physiological variables	Body weight (kg)	61.50 ± 8.62	−5.65 ± 12.90	68.17 ± 17.09	−1.74 ± 17.10	2.27 (0.007)
	Body fat (%)	32.01 ± 5.11	−2.52 ± 7.09	37.35 ± 5.85	1.41 ± 8.18	5.31 (<0.001)
	Fasting glucose (mg/L)	92.74 ± 6.76	−3.26 ± 16.88	103.32 ± 15.63	1.70 ± 22.93	1.351 (0.031)
	HbA1c (%)	5.35 ± 0.31	−0.18 ± 0.38	5.59 ± 0.34	0.04 ± 0.49	2.37 (0.019)
Diabetes knowledge		0.64 ± 0.93	0.11 ± 0.90	0.62 ± 0.89	0.10 ± 0.09	−0.54 (0.558)
Dietary habits		4.07 ± 0.30	0.74 ± 0.63	3.79 ± 0.43	0.34 ± 0.58	−3.63 (<0.001)
Health-promoting lifestyle profile	Health responsibility	3.51 ± 0.54	0.74 ± 0.97	2.75 ± 0.58	0.16 ± 0.82	−3.52 (0.001)
	Physical activity	3.50 ± 0.67	0.89 ± 0.91	2.81 ± 0.91	0.23 ± 0.91	−3.84 (<0.001)
	Nutrition	3.77 ± 0.58	1.52 ± 0.95	2.51 ± 0.73	0.14 ± 1.23	6.85 (<0.001)
	Spiritual growth	3.47 ± 0.72	1.23 ± 1.22	2.91 ± 0.51	0.68 ± 0.93	6.77 (<0.001)
	Interpersonal relationship	3.56 ± 0.56	1.19 ± 0.87	2.27 ± 1.08	0.02 ± 0.94	7.19 (<0.001)
	Stress management	2.19 ± 0.67	−0.42 ± 0.93	2.61 ± 0.90	−0.12 ± 0.95	−1.72 (0.088)
	Total	2.82 ± 0.32	0.34 ± 0.49	2.64 ± 0.38	0.18 ± 0.39	−11.18 (<0.001)
Parenting stress		3.30 ± 0.32	0.02 ± 0.60	3.52 ± 0.70	0.10 ± 0.81	0.71 (0.482)

## Data Availability

The data presented in this study are available on request from the corresponding author.

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
