# Peer review of "Self-Management Mobile Virtual Reality Program for Women with Gestational Diabetes"

_ijerph, 2021, doi:10.3390/ijerph18041539_

Round 1

Reviewer 1 Report

Interesting proposal.

The "quasi-experimental" approach worries me a bit. need to mention something about it in your conclusions to convince readers of the proposal's soundness.

Please check attached document

Author Response

We really appreciate your comment. Your comments will gear up our effort to conduct more significant studies in the future.

We have edited the manuscript point by point to reflect your advice.

Reviewer 2 Report

This is an interesting and challenging paper. The research is addressed to explore the potential contribution of Virtual Reality to prevent type 2 diabetes after childbirth in women previously diagnosed with a gestational diabetes mellitus during pregnancy

The higher value of this paper is the analysis of intervention policies using a mobile Virtual Reality self-management program in this field of health sciences. The program has been developed by the authors and include information about a neonatal first aid program, dietary recommendations, exercises to lose weight and stress relief activities.

This investigation is based on a quasi-experimental empirical and interventional study. The research design, the sample selection and research methods seem appropriate to provide significant evidences and fulfill the aims of the study. Although the sample is not very large (119 women) and it exclusively focuses on three South Korean hospitals, the findings shown in Table 2 are quite significant to infer a few conclusions about the favorable influence of this Virtual Reality program.

I would made only a few recommendations, to make clearer some aspects of the paper. The objective of the paper seems to be too ambiguous. The authors aim at intervention using a mobile VR self-management program, may be useful for postpartum management of gestational diabetic women because women are physically uncomfortable after childbirth and have difficulty visiting hospitals with responsibility for raising children. However, the absence of specific research questions is visible. Maybe the authors should specify their hypotheses, connected with the variables included in the empirical analysis.  

To implement quasi-experimental methods, the assignments have to ensure that both the experimental and control groups are equivalent. This severe restriction seems to be correctly solved, according to the characteristics displayed in Table 1. However, I am missing a more detailed information about some critical pieces of the research methods. In particular, information about the criteria used by the authors for the assignment to the treatment condition (control or experimental group) is unclear. Is it an experiment based on a random assignment?

If possible, a better understanding about the reasons of patients who decide to withdraw from the study could provide some interesting complementary information. Do they have some characteristics in common?

In my opinion, the different categories of items selected for the measurement are poorly rooted on previous findings. Apparently, they have been used once for the investigation of other pathologies. Are these variables also appropriate for the study of glucose intolerance? Maybe this point would deserve a short discussion in the paper.

Are the findings influenced by age, birth experience of family precedents in glucose disorders of patients? Exploration of these (potentially) hidden connections could enhance the value of the paper.

I am also worried about the internal validity of the results. Have the authors kept in mind that other possible reasons could influence these outcomes? In particular, the influence of environmental factors as differences in the type of occupation, location or educational level… If the authors are not able to gather this information, this limitation could be included in the Discussion section.

In addition, regarding to the external validity of the experiment it would be necessary to include a new limitation of the research: the extent to which the results obtained from this study can be generalized to other group of population or social context. Finally, the design of tables and figures could be improved to facilitate the reading of the paper.

Author Response

(The authors gave the same response as above.)

Round 2

Reviewer 1 Report

The document has been clearly improved, new information provided gives a clearer idea of the whole research. Congratulations